# Neuroprotective Ability of Apocynin Loaded Nanoparticles (APO-NPs) as NADPH Oxidase (NOX)-Mediated ROS Modulator for Hydrogen Peroxide-Induced Oxidative Neuronal Injuries

**DOI:** 10.3390/molecules26165011

**Published:** 2021-08-18

**Authors:** Manisha Singh, Shriya Agarwal, Raj Kumar Tiwari, Silpi Chanda, Kuldeep Singh, Prakhar Agarwal, Aishwarya Kashyap, Pranav Pancham, Shweta Mall, Rachana R., Shalini Sharma

**Affiliations:** 1Centre for Emerging Diseases (CFED), Department of Biotechnology, Jaypee Institute of Information Technology, Sector-62, Noida 201309, Uttar Pradesh, India; shriyaz.dd@gmail.com (S.A.); pranavpanchm@gmail.com (P.P.); rachana.dr@iitbombay.org (R.R.); 2Pharmacognosy and Phytochemistry, School of Health Sciences, Pharmaceutical Sciences, UPES, Dehradun 248007, Uttarakhand, India; raj_t365@yahoo.com; 3Pharmacognosy and Phytochemistry, IEC School of Pharmacy, IEC University, Solan 174103, Himachal Pradesh, India; only_shilpi@yahoo.com; 4Department of Chemistry, Maharishi Markandeshwar (Deemed to Be University), Mullana 133207, Haryana, India; singh@orgsyn.in; 5Department of Biosciences and Bioengineering, Indian Institute of Technology, Bombay 400076, Maharashtra, India; aprakhar50@gmail.com; 6Department of Biotechnology, Vellore Institute of Technology, School of Bio Sciences & Technology (SBST), Vellore 632014, Tamil Nadu, India; kashyapaishwarya19@gmail.com; 7Department of Animal Genetics and Breeding, Southern Regional Station of Indian Council of Agriculture Research—Research Institute, Bangalore 560030, Karnataka, India; shweta06.mall@gmail.com; 8Sunder Deep Pharmacy College, NH-9, Delhi-Meerut Expressway, Ghaziabad 201015, Uttar Pradesh, India; shalinia023@gmail.com

**Keywords:** reactive oxygen species (ROS), cytotoxicity, antioxidant activity (AO), transmission electron microscopy (TEM), release kinetics

## Abstract

Apocynin (APO) is a known multi-enzymatic complexed compound, employed as a viable NADPH oxidase (NOX) inhibitor, extensively used in both traditional and modern-day therapeutic strategies to combat neuronal disorders. However, its therapeutic efficacy is limited by lower solubility and lesser bioavailability; thus, a suitable nanocarrier system to overcome such limitations is needed. The present study is designed to fabricate APO-loaded polymeric nanoparticles (APO-NPs) to enhance its therapeutic efficacy and sustainability in the biological system. The optimized APO NPs in the study exhibited 103.6 ± 6.8 nm and −13.7 ± 0.43 mV of particle size and zeta potential, respectively, along with further confirmation by TEM. In addition, the antioxidant (AO) abilities quantified by DPPH and nitric oxide scavenging assays exhibited comparatively higher AO potential of APO-NPs than APO alone. An in-vitro release profile displayed a linear diffusion pattern of zero order kinetics for APO from the NPs, followed by its cytotoxicity evaluation on the PC12 cell line, which revealed minimal toxicity with higher cell viability, even after treatment with a stress inducer (H_2_O_2_). The stability of APO-NPs after six months showed minimal AO decline in comparison to APO only, indicating that the designed nano-formulation enhanced therapeutic efficacy for modulating NOX-mediated ROS generation.

## 1. Introduction

Cellular responses to oxidative stress in neuronal injuries have led to actualization of the hypothesis that inhibition of ROS (reactive oxygen species) formation is a comparatively stronger avenue to explore efficacious therapeutic possibilities for its treatment. By relating to this concept, earlier, the primary research in this area was more focused towards understanding the process of ROS production in intracellular pathways and for the identification of oxidases (mono/di oxygenase) in oxidative phosphorylation or peroxidases processes [1]. Furthermore, there were also reactive nitrogen species (RNS) like nitric oxide (NO^−^) and peroxynitrite (NO_3_^−^) that contributed equally to elevating oxidative stress levels in the central nervous system (CNS) particularly, or in the entire body largely. However, in recent times, NADPH oxidases (nicotinamide adenine dinucleotide phosphate hydrogen oxidase) have been marked prominently as an essential precursor for ROS production, specifically in neuronal cells, contributing towards initiating most of the pathological causes for chronic forms of neural degradation, including neurodegenerative disorders (NDDs) [2]. Moreover, nitrogen oxide (NOX) family protein homologs were also reported to be the only class of catalytic enzymes that primarily produces superoxide ions (superoxide anion, O_2_¯), hydrogen peroxide (H_2_O_2_), and hydroxyl radicals (OH^−^) altogether. This family consists of seven protein homolog subunits (NOX1, NOX2, NOX3, NOX4, NOX5 and dual oxidases—Duox1 and Duox2) of NADPH oxidase enzyme complex, wherein, NOX 1, 2, and 4 oxidases are highly expressed in human brain tissues and induces neuronal degradation. Normally in microglial cells, the cytosolic components, specifically p67-phox, p40-phox, and p47-phox, are not bound to NOX subunits. However, under stressed physiological conditions, when activation of G protein (Rac1 and Rac2) and phosphorylation of p47-phox takes place, these cytosolic components translocate to the transmembrane sites and bind with the cytochrome b588 membrane and activate NAPDH oxidase [3]. This process eventually leads to the generation of superoxide ions (O_2_¯) binding with FAD (Flavin adenine dinucleotide) and heme group of NOX subunits, further getting released across the membrane to increase ROS generation in cortical regions of the brain [4]. Consequently, in due course of time, the up-regulation of NOX-dependent ROS formation generates a superoxide anion that gets converted to hypohalous acids and per oxy nitrite in phagocytic cells; this too leads to neuronal damage. Thus, the increased interest in investigating the harmful effects of NADPH oxidase in neuronal degradation or injuries can only be prohibited by some largely known potential inhibitors of NADPH oxidase. Over the last decades, many research reports have identified Apocynin (APO) as the most suitable therapeutic compound to combat neuronal disorders. This is attained by halting the ROS generation in microglial cells and restricting the NOX2 overproduction with a fair possibility of the same in other neural cells [5]. Furthermore, APO prevents the binding of cytosolic components (p47-phox and p67-phox) of NADPH oxidase assembly and binds to the cytosolic component and inhibits its translocation or binding with membrane-associated components. Moreover, it has also been reported as a strong antioxidative agent in Phase I clinical trial studies and has proved to have established neuroprotective effects against ischemic injuries and stroke conditions [6]. However, APO has exhibited some potential pharmacokinetic limitations, including a higher rate of degradation, low bioavailability, and poor solubility concerns, restricting its efficacy [7]. Therefore, this study is focused on enhancing its therapeutic efficiency and encapsulation in a nanocarrier-based delivery system. This experimental exploration is extremely useful and could lead to effective utilization of its efficacy, along with the higher possibility of recovery from oxidative stress-related neural injuries.

## 2. Results

### 2.1. Optimisation and Statistical Analysis of Prepared APO-Loaded Nanoparticles (APO-NPs)

The process parameters involved in formulating APO-NPs were evaluated statistically by using two-level factorial design with Statease Design expert (Version 11) software [8]. After the optimization of various dependent and independent parameters of APO-NPs, the formulation was sorted statistically on the basis of APO encapsulation efficiency percentage (EE%) inside the NP matrix. Further, the results exhibited the possible runs for different optimized parameters, i.e., Chitosan (CS), Polycaprolactone (PCL), sonication time, and the rate of addition. The maximum EE % of 93.4% was observed in the 5th run whereas a minimum of 53.5% in the 2nd run (Table 1). Two-level factorial analysis was performed to evaluate the interaction between the optimized parameters and their optimal level response using 3D surface curve graphical representation (Figure 1), wherein the predicted versus experimental EE of the APO-NPs demonstrated a close comparison between the plotted predicted and experimental values, suggesting that all the optimized variables of the model were correlated with each other and validate the formulation of nanoparticles.

### 2.2. Particle Size (PSA) and Zeta Potential (ZP) Analysis

After fabrication of NPs, the next essential feature is to assess their size, dispersibility index, or average value (nm) of their size variability. A metric for size variability is the polydispersity index (PDI), a unit less quantity derived from the cumulate analysis and equivalent to the relative discrepancy of the distribution [9]. Particle size and size distribution are two essential touchstones of NPs and they influence other dependent factors such as drug release rate and biodistribution of therapeutic compound at the targeted site. In addition, the concerns related with size stability is another, more crucial factor for NPs than other drug delivery systems [10]. Therefore, based on EE (%), the highest EE exhibiting formulation (APO-NPs) was selected for the PSA and PDI analysis—103.6 ± 6.8 nm and 0.128 ± 0.08, respectively (Figure 2). Additionally, the measure of electric charge present on the NPs surface, also known as zeta potential (ZP), is another significant and considerable factor in deciding the magnitude of NPs stability in colloidal solution. Here, the ZP of the optimized APO-NPs was found to be −13.7 ± 0.43 mV (Figure 2).

### 2.3. FT-IR Analysis

Comparative FT-IR scan of APO, NPs, and APO-NPs was plotted to assess the presence of topographical constituents in the test samples. Here, the test samples were irradiated with IR radiations and the functional groups present on their surface were observed [11]. After scanning, the results exhibited signature peaks of APO at 3027.8 cm^−1^, suggesting the OH and CH bonds with carboxylic acid and alkene as functional groups, whereas the elevated peak ranges from 1703.8–1705.3 cm^−1^ showed C=O bond with carboxylic acid as a functional group. Similarly, the depressed peak ranged from 1387.3–1394.7 cm^−1^ exhibited C-X bond with fluoride as a functional group while other side ridges showed other various bonds like C-X bond, C=O bond and C-N bond with alcohols, amines, and fluoride as functional groups (Figure 3). In the case of NPs and APO-NPs, the signature peak ranges from 3383.7–3419.7 cm^−1^, showing an N-H bond with amines and amide as functional groups; the rest retained the same peaks and no additional peak structure was observed in the throughout scan of APO-NPs.

### 2.4. Transmission Electron Microscopy (TEM)

Transmission electron microscopy reveals the morphological structure and size of the nanoparticles [12]. The TEM imaging of the optimized APO NPs was taken at 10,000× magnification on a 100 nm scale. The morphological view of the APO-NPs was observed to be almost spherical with a size range of 15–90 ± 2.67 nm (Figure 4). This data was found in compliance with previous DLS data.

### 2.5. Physicochemical Parameters

The estimation of different physicochemical parameters of APO-NPs like pH, density, conductivity, and viscosity was measured. The pH of the formulation was observed to be 6.9 ± 0.63, suitable enough to cross any biological system. Further, the density and conductivity were found to be 0.968 g/mL and 7.86 mS/m, respectively, corresponding well with the existing properties of almost all biological systems. Thereafter, measured viscosity of the APO-NPs (0.967cP) (Table 2) exhibited good flowability and negligible resistance to the surface, proposing the suitability of the formulation for administration via most delivery routes and easily permeable through the biological barriers of human systems [13].

### 2.6. In Vitro Drug Release Kinetics Analysis

The APO release pattern from the NPs was studied using in vitro drug release kinetics modelling and it was observed that APO in its pure form followed a burst release effect in the receptor media, within 4 h of its addition into the donor compartment of the assembly. The data was repeated in triplicate and the average of the results obtained were plotted in the cumulative drug release (CDR) graph (Figure 5); the same set was also plotted in different release kinetics model equations to analyze the pattern of APO release from the NPs matrix. The result contrasted with the release profile shown by APO-NPs, wherein it followed a linear profile of diffusion through the dialysis membrane into the media. The slow rate of release is suggestive of a homogenous encapsulation of APO in the NPs [14,15] and it showed a maximum release of 91.52 ± 1.26% in 12 h (Figure 5). Subsequently, the compound release kinetics for APO-NPs was best explained by the zero order model of drug release kinetics, as the highest linearity (R^2^ = 0.985) was obtained for this model, followed by Hixon’s equation (R^2^ = 0.958), first order (R^2^ = 0.913), Higuchi’s equation (R^2^ = 0.909), and Korsmeyer—Peppas equation (R^2^ = 0.781).

### 2.7. In Vitro Cytotoxicity: Qualitative and Quantitative Analysis

#### 2.7.1. Fluorescence Microscopic

Cultured PC12 cell lines were subjected to viability assessment after inducing the oxidative stress condition by H_2_O_2_. This was done by ethidium bromide/acridine orange (EB/AO) staining, followed by visualizing the stained cells under fluorescent microscope, using fluorescein filter at 40x magnification. The EB/AO assay method is employed to check for cellular viability wherein AO permeates the viable cells, imparting a fluorescence green color with the intact cell structure, whereas EB crosses the cell membranes only when they show apoptotic changes transmitting a red/dark orange fluorescence color. The obtained images showed the dense cloud of dead cells in the H_2_O_2_ treated cells (with more nuclear fragmentation, cell membrane breakdown, and the plasma membrane blistering chromatin condensation (Figure 6B). However, in APO-treated cells, they showed early apoptotic appearances with denser green color and small orange appearances on the sides of the cells, suggesting the initiation of necrosis though lesser than the H_2_O_2_ treated ones (Figure 6D). In comparison to the control (Figure 6A), NPs (Figure 6C), and H_2_O_2_ set, APO-NPs were able to preserve better viability in cells, though they slightly appeared to be stressed but still holding most of their cellular framework intact (Figure 6E). Although cell density of both APO-NPs and extract (APO)-treated cells in terms of the number of nucleus visible was not as low as in the case of H_2_O_2_, the cellular disintegration seems to be less in APO-NPs, in comparison to the H_2_O_2_ treated cells.

#### 2.7.2. Quantitative Analysis—MTT Assay

As discussed earlier, the APO-NPs were tested on PC12 in triplicate and after analyzing the obtained results, it was observed that with the increase in the concentration of APO NPs samples, the percentage of cell viability increased to 103.2 ± 2.3% with the highest APO concentration (10 µg/mL) (Figure 7a) after 12 h. Again, the same pattern of remarkable increase in the count of viable cells after 24 h was noticed (117.93 ± 1.7%) at 10 µg/mL concentration (Figure 7b). Consecutively, APO alone also showed an increase in cell viability at both 12 h (105.2 ± 1.7%) and 24 h (108.37 ± 1.21%), suggesting that the compound itself is not toxic to the neural cell, rather was promoting cell growth. However, the formulated APO-NPs exhibited better compliance and growth than the APO, though the difference between the two was not appreciably significant. Furthermore, when the cells were subjected to the condition inducer (H_2_O_2_), it was noticed that there were not many significant toxic effects observed in the cells treated with APO-NPs. Comparatively, APO-NPs exhibited a slight decrease in cell viability (84.47 ± 0.76%) than the APO alone (58.21 ± 1.21%) after 12 h of exposure (Figure 7c). After exposure for 24 h with test samples (Figure 7d), the cell viability of APO was found to be 77.76 ± 1.39% in comparison to APO-NPs (90.3 ± 0.9%). The overall data indicated that the synthesized APO-NPs have negligible toxic effects on neural cells and their treatment in stressed conditions enabled the cells to hold their viability, in comparison to APO alone.

### 2.8. Antioxidant (AO) Analysis—DPPH and NO Assays

The AO analysis for all the test samples was done by DPPH and NO estimation. In the DPPH assay, it was observed that the scavenging activity of APO-NPs at 10 µg/mL was higher (89.7 ± 0.97%) in comparison to APO alone (51.31 ± 2.61%) and on the contrary, the percentage inhibition for NPs was 44.44 ± 1.48% and for AA, it was 96.6 ± 2.96% (Figure 8a). Similarly, in the NO assay, the calculated scavenging ability of the test samples (AA, NPs, APO, APO-NPs) was found to be 97.91 ± 1.92% (AA), 43.49 ± 1.83% (NPs), 58.74 ± 0.81% (APO), and 81.8 ± 1.3% (APO-NPs) (Figure 8b), suggesting that the AO of APO-NPs was better than the APO and more comparable to the positive control. Hence, the scavenging activity of the test samples can be ordered in the following manner: AA > APO-NPs > APO > NPs, confirming better quenching of nitrite ions in the samples. Therefore, from the collective results, it can be concurred that cumulatively, APO-NPs showed slightly better inhibition, in comparison to APO alone.

### 2.9. Stability Studies

A stability study was carried out to determine the sustenance of APO-NPs for longer periods. The optimized formulation (APO-NPs) samples were stored at 4 °C for six months and then analyzed for any degradation in antioxidative activity. The results of the antioxidative activity of the optimized formulation after six months is shown in Figure 9. The results show that the APO-loaded NPs formulation was more stable than the other test samples.

## 3. Discussion

As discussed earlier, APO is a known NADPH oxidase inhibitor that reflects its property to hold essential thiol groups of NADPH oxidase subunits together, thereby preventing the shift of the protein p47-phox sub-unit residing in the cytoplasm to the membrane [16]. Although it is a potential NADPH oxidase (NOX) mediated ROS modulator, due to its deficient aqueous solubility, reduced bioavailability, and fragile stability, its clinical utility and viability is restricted. Thus, we currently needed a turnkey solution by placing a suitable and appropriate nanocarrier system for APO that enhances its biopharmaceutical and pharmacokinetic profiles [17]. Moreover, a few nano formulation ideas with APO have already been applied, the inhibitor being an important phytocompound; for example, APO-loaded Bovine serum albumin (BSA) nanoparticles have successfully been prepared [18]. The shortcoming of this APO-based formulation is that it released 96% of APO from the NPs in 72 h, which is quite a slow process. However, in our study, we were able to overcome this by releasing 91.52 ± 1.26% of APO in approximately 9 h with zero-order release kinetics, suggesting that the rate of APO release was constant and sustained from the NPs lattice structure. In another research by Brenza et al., 2017 [19], antioxidant APO loaded polyanhydride nanoparticles targeting the mitochondria of neuronal cells was performed. Though it showed lesser encapsulation efficiency (43%), the sustained efficacious neuroprotection ability of the APO-Mito was reported to be quite high and effective in mitigating the oxidative stress-induced mitochondrial dysfunction and neuronal damage [20]. In our research study, the neuroprotective activity of the formulated and optimized APO-NPs was assessed on the PC12 cell line, and the results exhibited relatively stronger neuroprotective activity of APO-NPs in comparison to the APO alone. They also exhibited a suitable particle size range between 15–90 ± 2.67 nm with very narrow PDI score signifying a homogenous blend. Besides this, the formulation has not only enhanced the solubility and stability of APO itself, but its negative zeta potential charge may also help in improving the adsorption of plasma proteins. However, more elaborative pre-clinical research would be needed to understand the behavior of the prepared formulation inside the biological system [21].

## 4. Materials and Methods

Materials: Apocynin (APO), Chitosan (CS), Polycaprolactone (PCL) and 3-(4,5-Dimethylthiazol-2-yl)-2,5-diphenyltetrazolium (MTT), and Acridine Orange (A-6014) were procured from Sigma Aldrich, St. Louis, Burlington, VT, USA. Other chemicals were purchased from High Media Laboratories, Mumbai, Maharashtra, India: Ascorbic acid (AA), Griess reagent, DPPH reagent, Dulbecco’s Modified Eagle’s Medium (DMEM), and Dialysis membrane (Sigma D9652 with molecular weight cut-off of 14,000 Da). All other solvents and chemicals used were of HPLC and analytical grade, respectively.

### 4.1. Quantitative Analysis of Apocynin (APO) by the HPLC Method

The standard graph for APO was plotted by using reverse phase HPLC (Waters Alliance) apparatus (Model No. e2695 Waters, Vienna, Austria) equipped with a photodiode array detector (Waters, 2998-PDA) and auto sampler [22]. The system included an analytical reverse phase on a RP C18 column compartment (125 × 4 mm, 5 µm) with isocratic mode maintained at 27 ± 1.2 °C. The mobile phase consisted of acetonitrile, water, and acetic acid (60:40, *v*/*v*) blend and rate of flow was maintained at 1 mL/min with the injection volume of 20 µL of the APO sample (5 mg/mL) for 3 min [23]. The sample was then analyzed at 276 nm in triplicates.

### 4.2. Synthesis of APO-Loaded Polymeric Nanoparticles

#### Preparation of APO-NPs

The APO-NPs were synthesized by the ionic gelation method using chitosan (CS) and polycaprolactone (PCL) as the cross-linking oppositely charged polymers. The ionic gelation method is one of the recently distinguished techniques to prepare polymeric nanoparticles with hydrophilic drugs owing to its relative simplicity, efficiency, and better encapsulation efficiency (EE). In addition, CS is a naturally occurring polycationic polymer with glucosamine and N-acetyl glucosamine entities that blend well with anionic Poly (ε-caprolactone), forming an interfacial coacervate complexation by cross-linking [24] (as shown in Figure 10). For the preparation of APO-NPs, a CS (1–3%) solution with glacial acetic acid was prepared via constant overnight stirring at room temperature. On the other hand, PCL solution (0.1–0.3%) with glacial acetic acid was stirred until the solution became colorless. Then, APO (5 mg/mL) was added to the prepared CS solution and kept overnight for stirring. The PCL solution was added to the CS-APO mixture dropwise and the rate of addition for the same was maintained at 30 min [25,26]. Subsequently, the solution was kept under continuous stirring overnight to allow particles to stabilize in the solution mixture. Thereafter, the solution was centrifuged (11,200× *g*) for 20 min to separate the free compound (APO) molecules in the colloidal solution from the encapsulated NPs, again resuspending the pelleted APO-NPs in distilled water. The centrifugation washing step was repeated thrice to avoid any possibility of the unbound compound existing in the solution [27]. Later, validation of all the optimized formulations of APO-NPs was done by their encapsulation efficiency percentage (EE %), estimating the accurate amount of the drug delivered. The EE% was calculated by using (1):(1)$$EE\%=\frac{Wi-Wt}{Wi}*100$$
where, total drug concentration observed was denoted by *Wt* in the collected supernatant of the APO-NPs suspension and *Wi*, referred to as the total drug amount, which was initially added during the fabrication.

### 4.3. Statistical Optimization of APO-NPs

Statistical optimization of the formulated APO-NPs was verified by using the Stat-Ease Design Expert (version 11) software through a two-level factorial design. The statistical validation of all the fabricated NPs batches was done to determine the interaction between the various significant dependent/independent parameters and to attain the highest possible encapsulation efficiencies (EE %) for the same (Table 1). The two-level factorial analysis was performed to assess the interplay response between the optimized parameters with their optimal levels using 3D surface curve graphical representation (Figure 1). The predicted versus experimental EE of the APO-NPs demonstrated the nearest comparison amongst the predicted and experimental estimations, further highlighting the optimized variables of the model [28].

### 4.4. Characterization of the Optimized APO-NPs

#### 4.4.1. Particle Size Analysis (PSA) and Zeta Potential (ZP)

Particle size analysis (PSA) for the optimized APO-NPs was done by dynamic light scattering (DLS) method, wherein monochromatic light was passed through the particles, further creating fluctuation in the intensity of light [29]. The change in light intensity varied due to the Brownian movement of the particles in the colloidal sample and recorded the velocity of the Brownian motion to process the average size range of the particles through the Stokes-Einstein relationship. Subsequently, zeta potential (ZP) is marked as an electric potential inside the double layer interface present in the colloidal solution and it exhibits the ionic concentration potential gradient on the surface of the suspended particles. Both analyses were performed on Malvern Zeta sizer (Zetasizer Nano ZS) [30] and the sample preparation of the optimized nanoparticles (APO-NPs) were facilitated via sample dilution in the ratio of 1:50 in distilled water subsequently, followed by 20 min ultra-sonication, and subjecting them to PSA and ZP analysis.

#### 4.4.2. Fourier-Transform Infrared Spectroscopy (FT-IR)

Conventionally, several experimental studies have validated that the different chemical components tend to exhibit distinct absorption and emission pattern across an infrared spectrum. Furthermore, this attribute aids in locating as well as identifying different functional groups residing on the surface of the nanoparticles. For this, the FT-IR technique is used to find out the different molecular components of samples that involve transmission of infrared radiation for sample analysis. Employment of the potassium bromide (KBr) disc method was ensured to prepare all the test samples by blending them with KBr pellets, followed by sectioning it into thin film discs for grid analysis [31]. The prepared KBr pellet discs of the test samples were placed one by one inside the desiccator grid of the FT-IR (IR-810, JASCO, Tokyo, Japan) apparatus and all the residing functional groups were subjected to scanning at a band width from the 400–4000 cm^−1^ band for precise determination.

#### 4.4.3. Transmission Electron Microscopy (TEM)

TEM offers a comprehensive 2-D analysis of the surface conformation of the associated molecules and the resolution of the image obtained confirms the size range along with morphological features of the samples analyzed. The optimized samples (APO-NPs) were 100 times diluted in distilled water and ultrasonicated for 15 min before fixing them onto a carbon coated copper grid, especially treated with 2% phosphotungstic acid (PTA) [32]. This process of image analysis was performed by TEM (Hitachi, H-7500, Germany) at SAIF (Sophisticated Analytical Instrumentation Facility) lab, Punjab University, Chandigarh, Punjab, India, and the representative area imaging was recorded at a magnification of 10,000×.

#### 4.4.4. Physiochemical Parameters

The effect of different physicochemical parameters like pH, density, conductivity, and viscosity were also evaluated for APO-NPs. pH was measured using a digital pH meter (Mettler Toledo MP 220, Greifensee, Switzerland) at 37 °C, followed by density (E-Z Red SP101 Battery Hydrometer) and conductivity estimation (Orion Star A212) at 25 °C. Subsequently, the viscosity was determined using a viscometer (Brookfield), where the resistance of the NPs suspension was measured. All the tests were done in triplicate and the average value of the data was stated [33,34].

### 4.5. In Vitro Drug Release Kinetics Studies

The study of a compound release kinetic pattern from the encapsulated NPs state in the colloidal system is essential to assess the efficacy and therapeutic release of the formulated NPs to the targeted site of delivery [35]. The Franz diffusion cell assembly was used to evaluate the compound release from the formulated APO-NPs, where the assembly unit was kept on a constant magnetic stirrer at room temperature (37 ± 0.5 °C). The receiver compartment of the assembly was filled with phosphate buffer saline (18 mL, PBS pH 7.4) separated by a pre-treated dialysis membrane (D9652, Sigma Aldrich, Singapore) for the barrier base of media release. Then, 2 mL of the test samples (APO and APO-NPs) was added through a donor chamber individually and after a fixed interval of time (30 min) 1 mL samples were taken out from the receptor chamber with an equal amount of replacement for the equilibration. The absorbance of the withdrawn samples from the Franz diffusion unit was quantified by HPLC at 276 nm. The experiment was conducted for 12 h thrice and the average absorbance value of the extracted samples were plotted on the graph to record the cumulative drug release [36,37].

### 4.6. In Vitro Cytotoxicity Evaluation

Further, the neuroprotective potential of APO-NPs was assessed on the PC12 cell line that is derived from rat adrenal gland pheochromocytoma cells. Since these cell lines are used extensively to study the neurotoxicity, neuroprotection, and all other types of neuronal activities, we selected the same for our study. These cells were cultured by maintaining temperature at 37 °C and CO_2_ levels at 5%. The culture medium consisted of RPMI-1640 (Merck, Germany, cat. no. 8758), which was further supplemented with 5% FBS (fetal bovine serum) and 10% horse serum with 2 mM L-glutamine. To avoid contamination concerns in the cell line, a cocktail of 25 µg/mL gentamicin and 2.5 µg/mL amphotericin B was also added. We conducted cytotoxicity assessment by MTT assay in two different sets of experiments—in the first set without a stress inducer and in the other with a stress inducer—hydrogen peroxide (H_2_O_2_), which acted as an eminent ROS generator by disrupting the cell membrane, causing higher lipid peroxidation, along with DNA degradation in cellular functioning [38]. Subsequently, higher AO abilities of APO help in protecting the cells against deleterious oxidative stress induced by H_2_O_2_ by controlling ROS production. This analysis was done by both qualitative (ethidium bromide/acridine orange (EB/AO staining) and quantitative (MTT assay) methods [39].

#### 4.6.1. Fluorescence Microscopy Analysis—Acridine Orange/Ethidium Bromide (AO/EB) Dye

Visualization of the protective effects of various test samples (APO and APO-NPs) on PC12 cells was compared by subjecting them under AO/EB dye. In this double staining method, AO plays a significant role in cell-cycle studies as it can permeate through the cell and emit green fluorescence when bound to dsDNA by intercalating with the nucleic acid. In addition, being a metachromatic dye, acridine orange stains DNA green and RNA orange under appropriate conditions. However, EB imparts red/dark orange fluorescence when there is an altercation in cell membranes by intercalating inside the DNA [40]. Here, all the test samples were resuspended in 25 µL of a dye blend, formed by adding 100 µg/mL of each (AO/EB) dye in PBS (1 mL). Thereafter, 10 µL of the prepared sample was fixed on the slide with a coverslip for observation under a fluorescence microscope (Nikon Eclipse E 800).

#### 4.6.2. Quantitative Cell Viability Estimation—MTT Assay

To evaluate any cytotoxic effects of the optimized formulation, MTT (3-(4,5-dimethylthiazol-2-yl)-2,5-diphenyltetrazolium) assay was performed on a PC12 cell line, derived from adrenal medulla of the rodent model. MTT, a colorimetric assay, was performed, measuring the change in color from yellow to purple by reduction of MTT by mitochondrial succinate dehydrogenase into the mitochondria where it reduces into purple colored formazan, since reduction of MTT can be observed specifically in metabolically active cells confirming the count of viable cells [41].

For our experiment, we cultured the PC12 cells in RPMI2650 media (as mentioned above) and conducted passage by firstly washing it with PBS, followed by treatment with trypsin solution for 5 min at 37 °C, thereafter, centrifuging it at 1000× *g* for 5 min and resuspending the collected pellet in fresh medium, while discarding the supernatant. After passaging, they were seeded (triplicates) in 96-well plate at a concentration of 1 × 10^5^ cells per well in 96 well plates and incubated for 24 h at 37 °C for adherence to the new surface. After incubation, the media was removed and cells treated with 100 µL of different concentrations of the test samples (20–100 µg/mL) for 12 and 24 h, respectively, in the first set. In another set, after seeding the cells in a well plate and 24 h incubation, H_2_O_2_ (500 µM) [42] was added as a stress inducer and incubated for 4 min, Thereafter, the test samples were added and again incubated for 12 and 24 h. Further, after the incubation period, 10 µL of the MTT reagent was added to each well with 2 h of incubation, followed by addition of DMSO (0.1% *v*/*v*). Absorbance was measured at 570 nm in an ELISA plate reader and cell viability was calculated using Equation (2):(2)$$\%\;Cell\;viability=\frac{\mathrm{AT}-AB}{AC-AB}*100$$
where, AT refers to the optical density of all the varying test sample concentrations exposed to APO-NPs, whereas *AB* explains the optical density empty wells (without PC12 cells) and the optical density of the positive control is termed as *AC* [43].

### 4.7. Antioxidant (AO) Analysis

Antioxidant assays offer a comprehensive evaluation of the different aspects of antioxidant attributes exhibited by copious test samples; they were carried out by DPPH (2,2 diphenyl-1-picrylhydrazyl) and NO (nitric oxide) assay to determine the antioxidant activity of the tested samples.

#### 4.7.1. DPPH Assay

DPPH assay has applications in assessing the antioxidant activity (AO) of copious chemical compounds in response to free radical species. It shows reduction in the rate of chemical reactions, observed by change in color from deep violet to pale yellow after adding the DPPH solution. Here, the hydrogen donor is antioxidant in nature, quantifying free radical scavenger compounds [44,45]. The antioxidant effect is measured with the disappearance of DPPH (free radical) in test samples. 0.5 mL of the DPPH solution was added to 500 µL of the test samples prepared in different concentrations (10–50 µg/mL) and incubated for 30 min at 37 °C. The absorption maxima were recorded at 517 nm (purple) [46]. The scavenging activity (%) of the test sample was calculated by using (3):
(3)$$Scavenging\;activity\;\left(\%\right)=\frac{A_{OC}-\;A_{OS}}{A_{OC}}*100$$where, *A_OC_* and *A_OS_* refer to the absorbance of the control (ascorbic acid—AA) and the test samples, respectively, at 517 nm.

#### 4.7.2. Antioxidant (AO) Analysis—Nitric Oxide Assay (NO)

Several key biochemical as well as physiological mechanisms have been reported to regulate NO in human systems, including restoration of normal systemic circulation, propagation, and transduction of neuronal signaling, anti-cancerous, and antimicrobial activities. However, synthesis of Peroxynitrite anion (ONOO^−^) takes place when encountered with a free radical environment; thus, this expression of NO can guide further estimation of the NO radical quenching ability of a selected compound. AA was taken as a standard while conducting quantitative spectrophotometric analysis. 0.25 mL of 10 mM sodium nitroprusside solution was added to 1 mL of the test samples (APO, NPs, and APO-NPs) at different concentrations (10–50 µg/mL) and incubated for 3 h at 37 °C with 500 µL of Griess reagent and to this purple-colored chromophore (Griess Reagent) coupled with dihydrochloride, ethylenediamine, naphthyl was added for measuring absorbance at 546 nm [47]. Formation of this complex also acts as an indicator because reaction occurs due to diazotization of nitrite with sulphanilamide and to calculate percentage inhibition, using (4):
(4)$$\%\;Inhibition=\frac{I_{c}-{\;I}_{T}}{I_{c}}*100$$where, I_c_ refers to absorbance of the control and I_T_ to test samples, respectively, at 546 nm [31].

#### 4.7.3. Stability Studies of APO-NPs

A stability study was carried out to determine the stability of all the test samples (APO, NPs, and APO-NPs). These samples were stored at 4 °C for 1 year and then analyzed again to check for any sort of degradation in their antioxidative activity [48]. Comparative analysis of the antioxidant activity was measured by DPPH (2, 2-Diphenyl-1-picrylhydrazyl) and NO (Nitric Oxide) Assay with freshly prepared and 1-year stored samples [49].

### 4.8. Statistical Analysis

All the experimental datasets in the present study are reported as mean ± SD and their validation was done by two-way ANOVA with *p* value < 0.01.

## 5. Conclusions

The present research study highlights the formulation and optimization of APO-NPs by testing different concentration of polymers by the ionic gelation method. After optimization, it was found that 1% CS and 0.1% PCL are the best suited concentrations based on their encapsulation efficiency (93.43%). The average particle size of the prepared APO-NPs was recorded as 103.6 ± 6.8 nm with surface charge potential of −13.7 mV, which is quite suitable for crossing biological systems. Furthermore, the release kinetics data showed that APO was released from the formulation in a sustained release pattern and maximum of 97% of APO was released in the first 24 h. In addition, the cytotoxicity analysis of APO-NPs against PC12 cell lines exhibited higher neuroprotective effects comparatively, even after inducing a stressor, suggesting better sustainability than its natural form. The therapeutic potential of the APO-NPs was also tested for their AO abilities and the results confirmed that the quenching ability of the APO-NPs was better than that of the pure compound (APO) and significantly comparable to the standard (AA). In conclusion, we can confirm that the developed APO-loaded nano formulation will provide much improved clinical efficacy and limit all biopharmaceutical obstacles related to it.

## Figures and Tables

**Figure 1 molecules-26-05011-f001:**
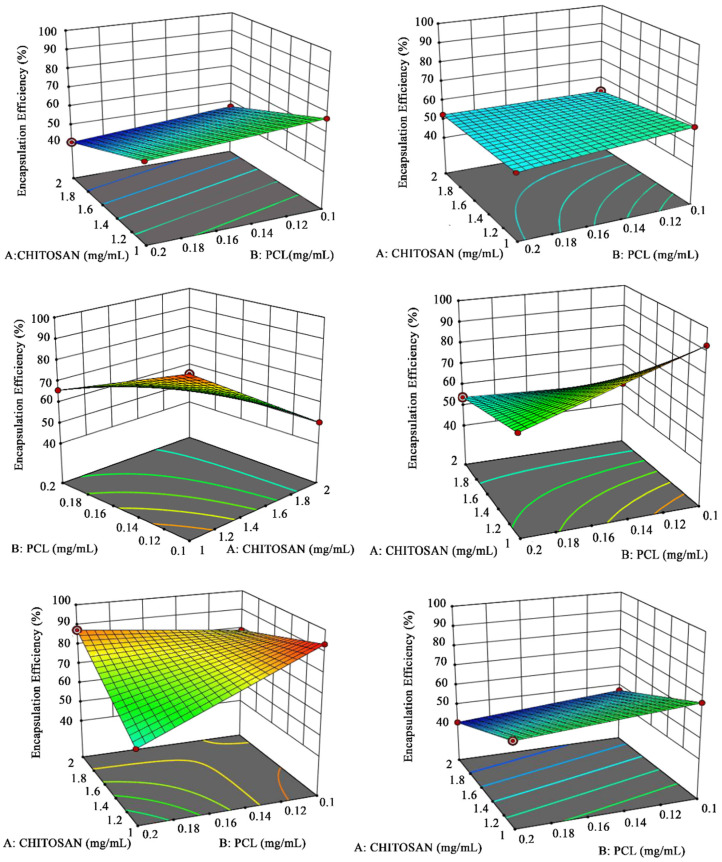
Response curve graphs representing the effects of interaction between the sonication time and rate of addition at the minimum and maximum concentration of CS and PCL.

**Figure 2 molecules-26-05011-f002:**
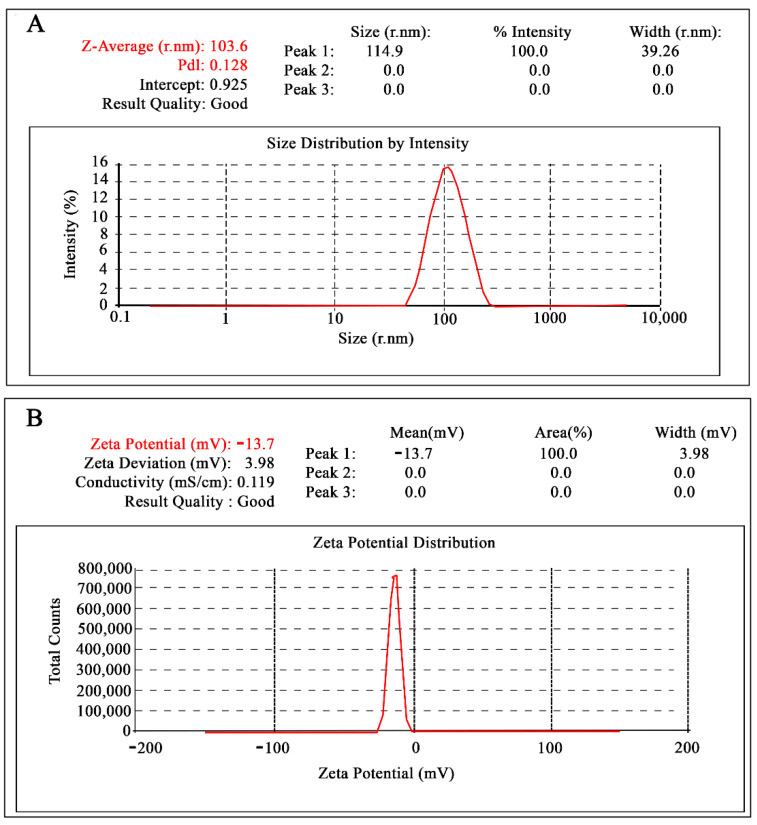
Graphical data showing particle size (**A**) and zeta potential (**B**) of the formulation for optimized APO-NPs.

**Figure 3 molecules-26-05011-f003:**
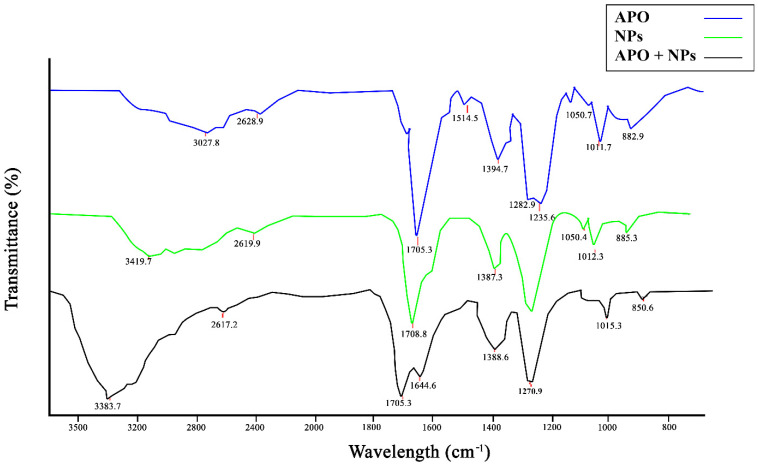
FTIR analysis graph scan of APO, NPs, and APO-NPs, respectively, from 400–4000 cm^−1^.

**Figure 4 molecules-26-05011-f004:**
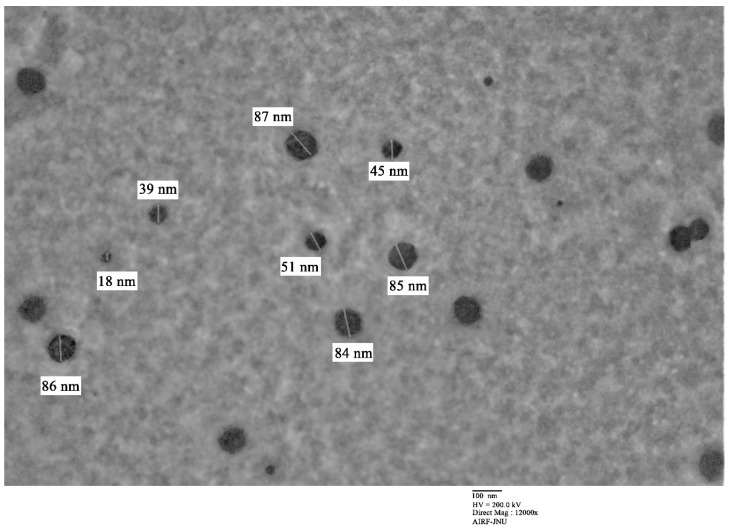
Micrograph showing TEM analysis of APO-NPs at a 100 nm scale of 10,000× magnification.

**Figure 5 molecules-26-05011-f005:**
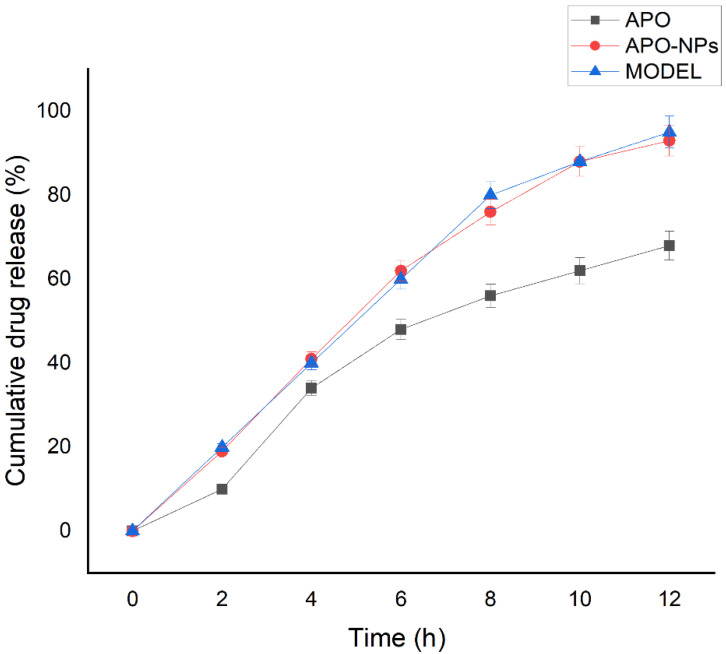
The comparative analysis of cumulative drug release (CDR %) of APO and APO-NPs.

**Figure 6 molecules-26-05011-f006:**
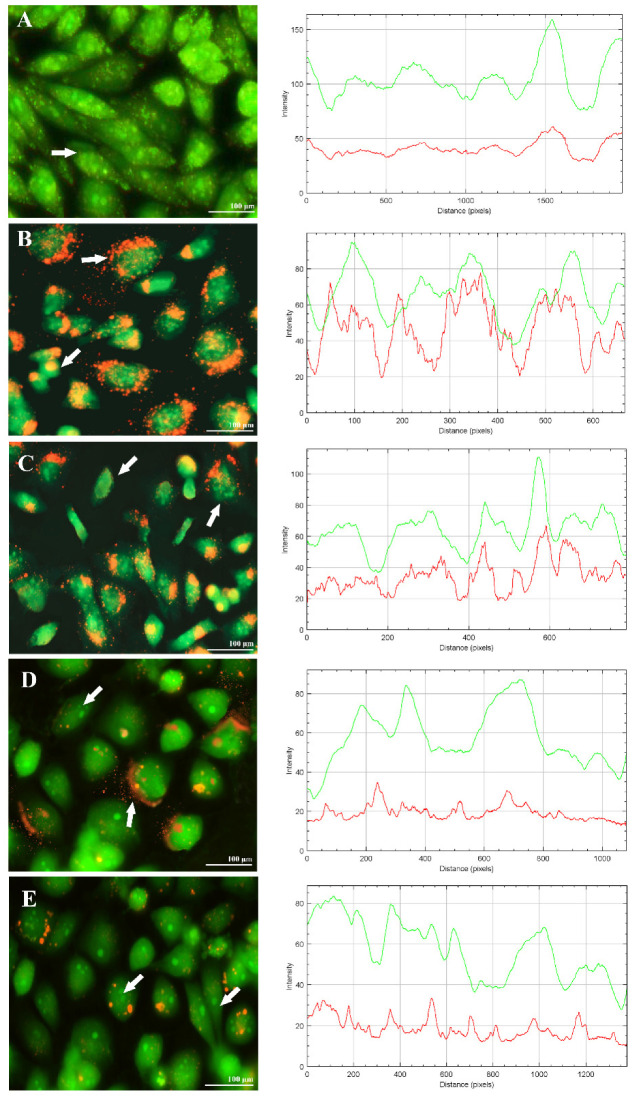
Fluorescence viewing (40X) of the PC12 cell line, after treating them with H_2_O_2_ along with various test samples for 12 h. Image Description: (**A**) Control (**B**) H_2_O_2_ (**C**) H_2_O_2_ with NPs (**D**), H_2_O_2_ with APO (**E**) H_2_O_2_ with APO-NPs.

**Figure 7 molecules-26-05011-f007:**
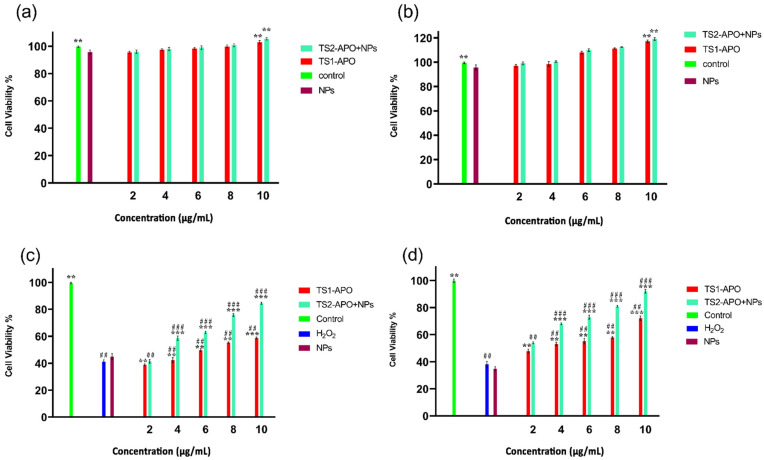
(**a**,**b**): Depicting comparative effect of APO-NPs on cell viability (%) against APO and the control at 12 and 24 h, respectively. Data is expressed in terms of mean ± SEM (*n = 3*), significant changes are given as ** *p* < 0.05 and *** *p* < 0.01 as compared to APO and control. The graph represented that APO-NPs showed better cell viability than the other comparative groups. However, (**c**,**d**) showed the comparative therapeutic effects of APO-NPs on cell viability (%) against hydrogen peroxide (H_2_O_2)_ induced neural stress condition in the PC12 cell line. Data is expressed in terms of mean ± SEM (*n = 3*), significant changes are given as ** *p* < 0.05 and *** *p* < 0.01, as compared to the control and APO; ## *p* < 0.05 and ### *p* < 0.01 compared to H_2_O_2_ treated groups. The graph here exhibited more significant recovery than the other comparative groups. Abbreviations: APO = Apocynin pure compound; APO-NPs = Apocynin loaded optimized chitosan—PCL nanoparticles; NPs = Only chitosan—PCL optimized nanoparticles without any compound; H_2_O_2_ = Hydrogen peroxide induced cells.

**Figure 8 molecules-26-05011-f008:**
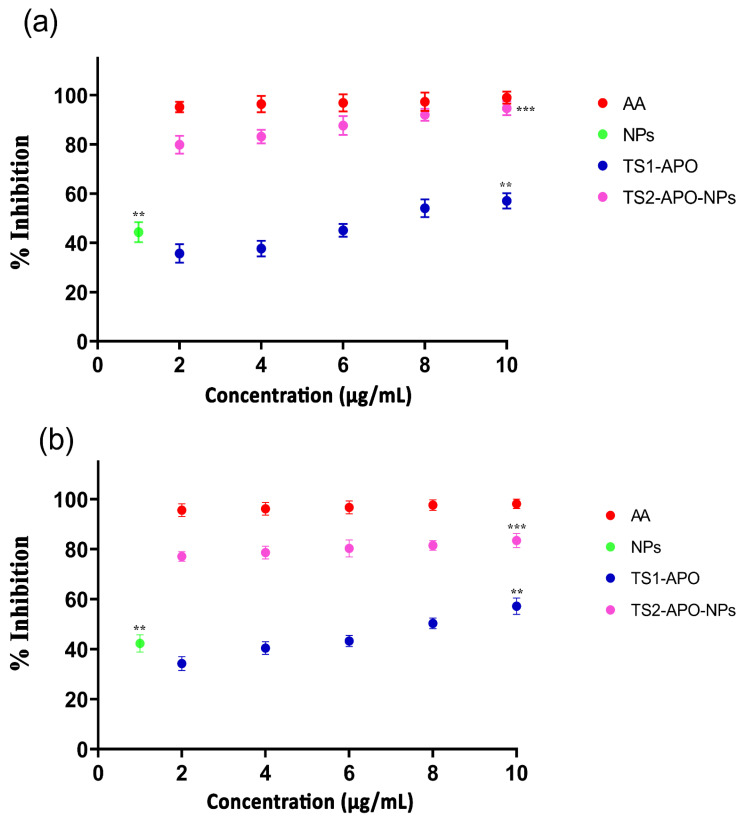
(**a**,**b**): The AO graph represents a comparative effect of APO-NPs on antioxidant ability against APO and the control (Ascorbic acid—AA) at 12 and 24 h, respectively. Data is expressed in terms of mean ± SEM (n = 3), significant changes are given as ** *p* < 0.05 and *** *p* < 0.01, as compared to APO and the control. The graph showed that APO-NPs have better antioxidant ability than the other comparative groups. Abbreviations: APO = Apocynin pure compound; APO-NPs = Apocynin loaded optimized chitosan—PCL nanoparticles; NPs = Only chitosan—PCL optimized nanoparticles without any compound; AA = Ascorbic acid as positive control.

**Figure 9 molecules-26-05011-f009:**
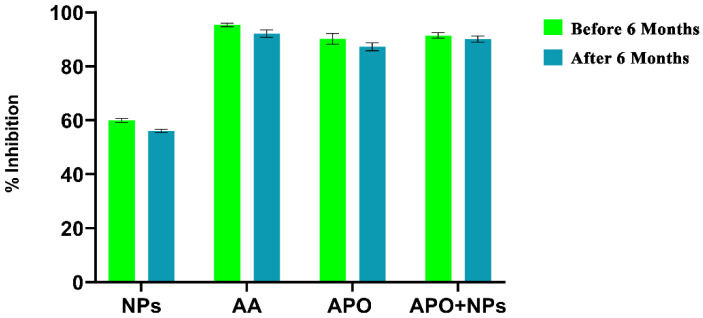
Comparative study by antioxidative analysis of the pure compound, NPs, and APO-loaded NPs using DPPH Assay.

**Figure 10 molecules-26-05011-f010:**
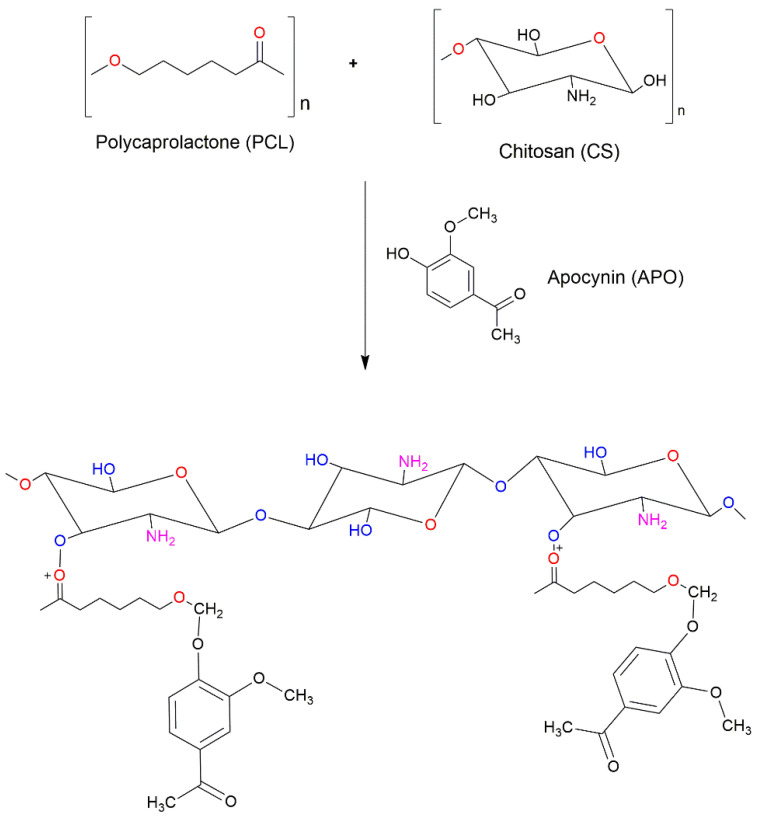
Lattice structure depicting the suggestive encapsulation of APO over the CS and PCL polymeric chain of NPs.

**Table 1 molecules-26-05011-t001:** Compiled data of all the statistical runs performed for all the formulated batches of APO-NPs under 2^4^ factorial designs.

Std	Run	A. CS (mg/mL)	B. PCL (mg/mL)	C. Sonication Time (min)	D. Rate of Addition (mL/min)	EE (%)
5	1	1	0.1	10	30	92.4
3	2	1	0.2	5	30	53.5
4	3	2	0.2	5	30	52.5
14	4	2	0.1	10	60	79.0
13	5	1	0.1	10	60	93.4
9	6	1	0.1	5	60	67.2
7	7	1	0.2	10	30	66.0
2	8	2	0.1	5	30	52.5
12	9	2	0.2	5	60	40.4
11	10	1	0.2	5	60	61.2
8	11	2	0.2	10	30	54.0
10	12	2	0.1	5	60	47.0
1	13	1	0.1	5	30	60.2
16	14	2	0.2	10	60	87.3
15	15	1	0.2	10	60	56.3
6	16	2	0.1	10	30	50.5

**Table 2 molecules-26-05011-t002:** Physicochemical parameters of APO-NPs.

Parameters	APO-NPs
Viscosity	0.967 cP
pH	6.9
Density	0.968 g/mL
Conductivity	7.86 mS/m

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
