# Peer review of "Neuroprotective Ability of Apocynin Loaded Nanoparticles (APO-NPs) as NADPH Oxidase (NOX)-Mediated ROS Modulator for Hydrogen Peroxide-Induced Oxidative Neuronal Injuries"

_molecules, 2021, doi:10.3390/molecules26165011_

Round 1
Reviewer 1 Report
In this paper, the generation of Apocynin loaded polymeric nanoparticles has been described, together with a preliminary characterization of their biological effects. Although the study is interesting, the biological part is poor in the methods and descriptions of results and citations are often not applicable. English quality is also an issue. For these reasons, in my opinion, the manuscript is far from publishing in Molecules at this point.
Below the detailed concerns.
Major points:
- Statistical analysis of experiments reported in figures 8, 9 and 10 is missing, even in the materials and methods section. A statistical analysis is necessary in order to compare the differences between the two groups of samples. The authors reported differences in the biological activity of APO-NP versus APO, which seems to be true for cell viability but not for antioxidant activity (85.7% versus 83.31%, in DPPH assay). Furthermore, in the description of results reported in figure 9 B, the authors stated the following percentages 65.74 ± 0.81% (APO) and 81.8 ± 1.3% (APO–NPs), thus concluding that antioxidant activity of APO - NPs is better than that of APO. However, the column height seems to not match with this description, suggesting instead no differences between the groups.
- In the same figures (8, 9 and 10) it is not mentioned if error bars represent standard deviation or standard error.
- Cell culture method is missing. Moreover, it is not reported, in the evaluation of biological activity, the number of experiments performed and the number of samples for each experiment (with the exception of triplicate for MTT test).
- Figure 7 seems to be inconclusive. Acridine-orange staining is not useful for cell viability (MTT is more appropriate). For morphology analysis, bright field images would have been more appropriate, coupled to the measurement of morphological parameters (i.e. cell body size). Why orange is so predominant in cells treated with APO or APO-NP? The images are completely different from both Control and H2O2 pictures: DNA versus RNA? Acidic vacuoles? Cell cycle? Contamination? Please describe and explain.
- Scale bar is missing in figure 7.
- What concentration of H2O2 was used?
- The authors reported that 100 μL of 1x103 cells per mL per well were seeded in a 96 wells plate, which should correspond to 100 cells per well. It seems a too low amount, considering also that cells were kept in culture for no longer that 24 hours. MTT signal in this condition would have been very low. Also, cell density in figure7, although 40X magnification, seems to not match with the number of plated cells.
- Bibliography is not pertinent, or at least not well placed. It must be all revised. Striking examples: 1, 2, 10, 33, 34, 35, 36, 37.
Minor points:
- Some paragraphs of the Results section should be moved to the m&m section (i.e. 2.1 and 2.2).
- Several typos, some examples: line 69-71, 93, 119, 243, 293, 422, 432, 382.
Author Response
REVIEWER - 1
Comment 1:
In this paper, the generation of Apocynin loaded polymeric nanoparticles has been described, together with a preliminary characterization of their biological effects. Although the study is interesting, the biological part is poor in the methods and descriptions of results and citations are often not applicable.
Answer to comment 1:
As per the suggestion, we have explained the biological concept in more detail in method and result sections.
Comment 2:
English quality is also an issue.
Answer to comment 2:
As per the suggestion, we have done the professional language editing and improvised the full manuscript.
Major points
Comment 3:
Statistical analysis of experiments reported in figures 8, 9 and 10 is missing, even in the materials and methods section. A statistical analysis is necessary to compare the differences between the two groups of samples.
Answer to comment 3:
We had done the statistical validation of all the data set with 2-way ANOVA and now we have mentioned the same in the draft as a separate section (4.8) and individually too in all content.
Comment 4:
The authors reported differences in the biological activity of APO-NP versus APO, which seems to be true for cell viability but not for antioxidant activity (85.7% versus 83.31%, in DPPH assay).
Answer to comment 4:
We have repeated this experiment and have plotted the graph again.
Comment 5:
Furthermore, in the description of results reported in figure 9 B, the authors stated the following percentages 65.74 ± 0.81% (APO) and 81.8 ± 1.3% (APO–NPs), thus concluding that antioxidant activity of APO - NPs is better than that of APO. However, the column height seems to not match with this description, suggesting instead no differences between the groups.
Answer to comment 5:
We have repeated this experiment and have plotted the graph again.
Comment 6:
In the same figures (8, 9 and 10) it is not mentioned if error bars represent standard deviation or standard error.
Answer to comment 6:
We have repeated this experiment and have plotted the graph again and, in the graph, we have put the error bars that represents the standard deviation.
Comment 7:
Cell culture method is missing. Moreover, it is not reported, in the evaluation of biological activity, the number of experiments performed and the number of samples for each experiment (with the exception of triplicate for MTT test).
Answer to comment 7:
We have now added the cell culturing details and deeper analytical evaluation of the biological activity in all the data sections. Also, we have added the count of experiments repeated in all sections.
Comment 8:
Figure 7 seems to be inconclusive. Acridine-orange staining is not useful for cell viability (MTT is more appropriate).
Answer to comment 8:
As per the query raised by the reviewer’s, we wish to apologised for the mistake that we made by only stating the qualitative assay to be acridine orange (AO) based but actually it was Acridine orange/ethidium bromide double staining assay which is a very common rapid assay performed to investigate the apoptotic damage from any compound in plant as well as in animal cells. Also, there was a wrong marking in the images which we have corrected now and we have added one additional image from our experiment to compare all the test samples in equivalence with MTT assay.
Acridine orange/ethidium bromide (AO/EB) double staining test is fast and statistically significant fluorescence method to determine the
(i). Morphological changes in nuclei structures,
(ii). Assessing the apoptotic insults,
(iii). Easy method to determine the cell viability and
(iv). To distinguish the viable apoptotic and necrotic cells from early and late stages.
This Stain is a cell viability dye that causes viable nucleated cells to attain fluoresce green and nonviable nucleated cells to fluoresce red/dark orange. And is equally used to assess the cell viability with the automated fluorescence cell counters.
Comment 9:
For morphology analysis, bright field images would have been more appropriate, coupled to the measurement of morphological parameters (i.e., cell body size). Why orange is so predominant in cells treated with APO or APO-NP? The images are completely different from both Control and H2O2 pictures: DNA versus RNA? Acidic vacuoles? Cell cycle? Contamination? Please describe and explain.
Answer to comment 9:
We wish to mention that the bright field images didn’t showed the proper demarcation and structural accuracy of the cell structure, therefore we decided to go for AO/EB staining assay wherein Acridine orange permeated the viable cells and binds to the nucleic acids. Binding to dsDNA causes acridine orange to fluoresce green and binding to ssDNA or RNA causes it to fluoresce red. On the other hand, Ethidium bromide emits red fluorescence by intercalation into DNA, when cells have altered the cell membranes.
Comment 10:
Scale bar on both axis X-axis and Y-axis are marked.
Answer to comment 10:
Comment 11:
What concentration of H2O2 was used?
Answer to comment 11:
We have used the 500µM concentration of H2O2 in the experiment and have put a reference for the same too in the manuscript. Please refer.
Comment 12:
The authors reported that 100 μL of 1x103 cells per mL per well were seeded in a 96 wells plate, which should correspond to 100 cells per well. It seems a too low amount, considering also that cells were kept in culture for no longer that 24 hours. MTT signal in this condition would have been very low. Also, cell density in figure7, although 40X magnification, seems to not match with the number of plated cells.
Answer to comment 12:
Thanks for the correction, we have by mistake referred it as 1x103 cells per mL, instead it should be 1x103 cells per µL. Now we have corrected the content with appropriate concentration volume.
Comment 13:
Bibliography is not pertinent, or at least not well placed. It must be all revised. Striking examples: 1, 2, 10, 33, 34, 35, 36, 37.
Answer to comment 13:
The correction is done in the bibliography.
Minor points
Comment 14:
Some paragraphs of the Results section should be moved to the m&m section (i.e. 2.1 and 2.2).
Answer to comment 14:
We have moved the contents to the material and methods section.
Comment 15:
Several typos, some examples: line 69-71, 93, 119, 243, 293, 422, 432, 382.
Answer to comment 15:
We have corrected the typos in the overall draft.
Reviewer 2 Report
The manuscript entitled “Neuroprotective ability of Apocynin Loaded Nanoparticles 2 (APO-NPs) as NADPH Oxidase (NOX)-Mediated ROS Modulator for Hydrogen Peroxide-Induced Oxidative Neuronal Injuries” by Shing and colleagues is aimed to develop APO-loaded polymeric nanoparticles (APO-NPs). Upon characterization of optimized APO-NPs by determining particle size and zeta potential along with TEM analysis, the antioxidant ability was quantified by DPPH and nitric oxide scavenging assays and in-vitro release profile. Cytotoxicity was evaluated on PC12 cells and cell viability was investigated after treatment with a stress inducer (H2O2). Finally, the stability of APO-NPs was checked after six months. Taken together, results indicate that the designed nano formulation has enhanced therapeutic efficacy when compared with APO alone for modulating NOX-mediated ROS generation.
Major concerns:
Extensive English revision.
Avoid repetitions (e.g. “eventually” is used 2 times in line 68)
H2O2 dose is not mentioned.
Acridine orange is not usually described as a dye to evaluate cell viability. A description should be provided, together with an explanation of the green and orange staining in the different experimental conditions and its correlation with cell viability.
In Fig. 8 and 9, statistical differences are not indicated.
Results from Fig. 8 are surprising. After 24 h, NPs alone (dose not mentioned) significantly decrease cell viability by more than 50%, but with APO-NPs are added to H2O2-treated cells, cell viability is increased. Does APO binding to NPs prevent their toxicity?
In Fig. 9, no differences between cells treated with APO alone or with APO-NPs are observed, contradicting the conclusions of the authors. Similarly, the authors conclude that the stability of APO-NPs is higher than that of other treatments, which is not shown in Fig. 10.
Author Response
REVIEWER - 2
The manuscript entitled “Neuroprotective ability of Apocynin Loaded Nanoparticles 2 (APO-NPs) as NADPH Oxidase (NOX)-Mediated ROS Modulator for Hydrogen Peroxide-Induced Oxidative Neuronal Injuries” by Singh and colleagues is aimed to develop APO-loaded polymeric nanoparticles (APO-NPs). Upon characterization of optimized APO-NPs by determining particle size and zeta potential along with TEM analysis, the antioxidant ability was quantified by DPPH and nitric oxide scavenging assays and in-vitro release profile. Cytotoxicity was evaluated on PC12 cells and cell viability was investigated after treatment with a stress inducer (H2O2). Finally, the stability of APO-NPs was checked after six months. Taken together, results indicate that the designed nano formulation has enhanced therapeutic efficacy when compared with APO alone for modulating NOX-mediated ROS generation.
Major concerns
Comment 1:
Extensive English revision.
Answer to comment 1:
As per the suggestion, we have done the professional language editing and improvised the full manuscript.
Comment 2:
Avoid repetitions (e.g., “eventually” is used 2 times in line 68)
Answer to comment 2:
We have corrected all such repetitions throughout the draft.
Comment 3:
H2O2 dose is not mentioned.
Answer to comment 3:
We have used the 500µM concentration of H2O2 in the experiment and have put a reference for the same too in the manuscript. Please refer.
Comment 4:
Acridine orange is not usually described as a dye to evaluate cell viability. A description should be provided, together with an explanation of the green and orange staining in the different experimental conditions and its correlation with cell viability.
Answer to comment 4:
As per the query raised by the reviewer’s, we wish to apologised for the mistake that we made by only stating the qualitative assay to be acridine orange (AO) based but actually it was Acridine orange/ethidium bromide double staining assay which is a very common rapid assay performed to investigate the apoptotic damage from any compound in plant as well as in animal cells.
Acridine orange/ethidium bromide (AO/EB) double staining test is fast and statistically significant fluorescence method to determine the
(i). Morphological changes in nuclei structures,
(ii). Assessing the apoptotic insults,
(iii). Easy method to determine the cell viability and
(iv). To distinguish the viable apoptotic and necrotic cells from early and late stages.
This Stain is a cell viability dye that causes viable nucleated cells to attain fluoresce green and nonviable nucleated cells to fluoresce red/dark orange. And is equally used to assess the cell viability with the automated fluorescence cell counters.
Comment 5:
In Fig. 8 and 9, statistical differences are not indicated.
Answer to comment 5:
We have corrected the figure – 8 and 9 along with statistical difference. Please refer to the content in manuscript.
Comment 6:
Results from Fig. 8 are surprising. After 24 h, NPs alone (dose not mentioned) significantly decrease cell viability by more than 50%, but with APO-NPs are added to H2O2-treated cells, cell viability is increased. Does APO binding to NPs prevent their toxicity?
Answer to comment 6:
The mentioned figures represent the condition after a stress inducer is introduced in the cell culture, now in this condition though NP do have its own AO property but that is not sufficient enough to combat with the stressed effects of the H2O2 alone therefore, minimal reversal action is initiated by the NPs thus, cell viability is calculated to be significantly decreased. However, APO is a known neuroprotective agent and when it is added in the NP matrix and incubated with the cells it exhibited the protection and restoration of the cells in the culture. So definitely encapsulation of APO in the NP matrix and its slow release from the same reduces the toxic effects of H2O2.
Comment 7:
In Fig. 9, no differences between cells treated with APO alone or with APO-NPs are observed, contradicting the conclusions of the authors. Similarly, the authors conclude that the stability of APO-NPs is higher than that of other treatments, which is not shown in Fig. 10.
Answer to comment 7
As per our study APO alone also showed the increase in cell viability at both 12 hours (105.2 ± 1.7%) and 24 hours (108.37 ± 1.21%), suggesting that APO itself is not toxic to the neural cells rather it was promoting the cell growth. Comparatively APO-NPs exhibited a slight decrease in cell viability (84.47 ± 0.76%) than the APO alone (58.21 ± 1.21%) after 12 hours of exposure. Then followed by the exposure for 24 hours with test samples, the cell viability of APO was 77.76 ± 1.39% in comparison to APO-NPs (90.3 ± 0.9%). The overall data indicated that the synthesized APO-NPs have negligible toxic effect on the neural cells and its treatment in stressed condition supported the cells to hold their viability back in comparison to APO alone.
Also, when statistical analysis was performed for the stability testing of the test samples after 6-months. It was observed that the decrease in the stability of the APO-NPs was very less in comparison to APO alone. Thus, we can say that the stability of APO-NPs (on storage) is more than that of APO individually.
Round 2
Reviewer 1 Report
The authors have revised the manuscript according to reviewer’ suggestions. English has been largely improved and most of the critical points have been addressed. However, there are still some issues to clarify before accepting the manuscript for publication. The revised form of the manuscript seems improved.
English now more suitable for publication.
- Again, statistical analysis of experiments reported in figures 8, 9 and 10 is missing. The authors specified that a two-way ANOVA was performed, without any indications of ANOVA data in the results, figure or figure legend. Also, post-hoc test is needed. Otherwise, it is impossible for the reader to understand data reliability and differences between groups.
- Cell culture method Culture conditions, medium, etc) is still missing.
- Bar is missing in figure 6.
- Check again all bibliography, a lot of not pertinent citations.
Author Response
Comments and Suggestions for Authors
The authors have revised the manuscript according to reviewer’ suggestions. English has been largely improved and most of the critical points have been addressed. However, there are still some issues to clarify before accepting the manuscript for publication. The revised form of the manuscript seems improved.
English now more suitable for publication.
Q1. Again, statistical analysis of experiments reported in figures 8, 9 and 10 is missing. The authors specified that a two-way ANOVA was performed, without any indications of ANOVA data in the results, figure or figure legend. Also, post-hoc test is needed. Otherwise, it is impossible for the reader to understand data reliability and differences between groups.
A1. The figures have been corrected and inserted and uploaded in the draft again.
Q2. Cell culture method Culture conditions, medium, etc) is still missing.
A2. The content has been corrected as per the review comment in the uploaded draft.
Q3. Bar is missing in figure 6.
A3. The figure has been redrawn and critically improvised as per the query.
Q4. Check again all bibliography, a lot of not pertinent citations.
A4. All the references have been checked again and re-done.

Reviewer 2 Report
The authors made substantial changes to the manuscript that made it clearer and with better quality. Although it has changed significantly, English can still be improved to make reading and understanding easier. Fig. 9 must be modified (font type, width and color of bars, etc.) in order to have an identical appearance to the other figures. Regarding the scientific content, the quantification of the number of apoptotic cells (Fig. 6) must be presented.
Author Response
Comments and Suggestions for Authors
Q1. The authors made substantial changes to the manuscript that made it clearer and with better quality. Although it has changed significantly, English can still be improved to make reading and understanding easier. Fig. 9 must be modified (font type, width and color of bars, etc.) in order to have an identical appearance to the other figures. Regarding the scientific content, the quantification of the number of apoptotic cells (Fig. 6) must be presented.
A1. Both the figure 6 and figure 9 has been redrawn and corrected as per the comment by the reviewer. The figures and graph has been improvised and inserted in the final uploaded copy of the manuscript along with the improvised figure legends.
